# Assessing the multifunctionality of service crops in mediterranean vineyards using a functional trait approach

**Léo Garcia**[1]*, **Aurélie Metay**[1], **Gaëlle Damour**[2,3]*

**1** ABSys, Univ Montpellier, CIHEAM-IAMM, CIRAD, INRAE, Institut Agro, Montpellier, France, **2** CIRAD, UPR GECO, Montpellier, France, **3** GECO, Univ Montpellier, CIRAD, Montpellier, France

* gaelle.damour@cirad.fr (GD); leo.garcia@institut-agro.fr (LG)

## Abstract

Vineyard soils face various agronomic issues such as poor organic carbon levels, erosion, fertility losses, and numerous studies have highlighted the ability of service crops to address these issues. Because biodiversity enhances the multifunctionality of managed ecosystems, service crop mixtures that increase functional diversity represent a promising option to improve vineyard sustainability. Plant functional traits play a crucial role in understanding ecosystem functions, serving as drivers for ecosystem processes and influencing ecosystem services, but the relationship between plant functional traits and ecosystem services is also complex. This study aimed to identify the links between the functional structure of the service crops associated with grapevines, the function they deliver and ecosystem function multifunctionality (EFM), in a Mediterranean vineyard. Thirteen different monocultures of service crop species were sown in the inter-rows of plots of 30 m length that covered one row and the two adjacent inter-rows, at random locations. We then studied 38 plant communities each composed of one of the sown service crop and the spontaneous vegetation that developed with it. At vine budburst, we simultaneously measured five indicators of ecosystem functions (runoff reduction, soil stabilization, soil mineral nitrogen supply for the vine, soil water supply for the vine, and community biomass production), along with 12 above- and below-ground functional markers of the community associated with these functions, in each plant community. Relationships between ecosystem functions and functional markers were analyzed by combining PCA, correlations and multiple linear regressions. We showed that traits upscaled at the community level (CWM) explained part of the targeted functions: significant correlations between traits and functions ranged from 0.33 to 0.6; the $R^2$ values of the linear regression models between functional indicators and the PCA axes derived from the traits ranged from 0.16 to 0.56. Additionally, we identified tradeoffs between functions, and observed that the biomass production was a major driver of soil-based ecosystem functions. In conclusion, functionally different communities provided different levels of functions

**Data availability statement:** All data files related to this article are available from the Dataverse depository with the doi: doi:10.18167/DVN1/0TKB41. URL: https://dataverse.cirad.fr/dataset.xhtml?persistentId=doi:10.18167/DVN1/0TKB41.

**Funding:** LG, AM: FertilCrop project, FP7 ERA-Net program CORE Organic Plus. The funders had no role in study design, data collection and analysis, decision to publish, or preparation of the manuscript.

**Competing interests:** The authors have declared that no competing interests exist.

and EFM. Designing service crops communities with complementary plant traits may be particularly relevant for increasing multifunctionality and agrosystem sustainability.

## Introduction

Many recent studies support the notion that diversified cropping systems can maintain high productivity levels while requiring fewer external inputs and resulting in lower externalities, compared to more simplified systems [1–3]; for example, Marini et al. [3] estimated annual yield increases of approximately 860 kg ha$^{-1}$ for winter cereals and 390 kg ha$^{-1}$ for spring cereals thanks to crop diversification, respectively, representing a 20–25% yield gain. These studies are based on the hypothesis of a positive relationship between biological diversity and the functions of ecosystems like biomass production, and nutrient cycling [4,5]. Diversification in cropping systems has been shown to enhance the buffering capacity and adaptive potential to increasingly frequent extreme climatic events driven by climate change [6]. Modification of the microclimate is one of the reasons, but the ecosystem functions that arise from diversifying crops, such as soil quality and fertility enhancements, can also help to buffer these systems against weather variability associated with climate change [7–10]. Analyzing the relationships between cropping system biodiversity and the functions it delivers is needed to evaluate the potential benefits of diversification and encourage stakeholders to shift to diversified, multi-service systems [11].

Plant functional traits play a crucial role in understanding ecosystem functions, serving as drivers for ecosystem processes and influencing ecosystem services [12–14]. These traits, responding to environmental changes, affect various processes such as soil retention, plant growth, water transport, and soil fertility [15–17]. The specific leaf area (SLA) is related to the potential growth rate of a species and is often used for trait-based approaches in agrosystems [18–20]. The leaf dry matter content (LDMC) is related to leaf fibers and plant decomposability and thus leaf mulch persistence [21–23]. In addition, the plant dry matter content (DMC) also relates to the persistence of soil cover after the mechanical termination of the whole plant. The leaf nitrogen content is also a marker of the acquisitive strategy of species and relates to the litter decomposition and mineralization [16,19,20,24]. Root traits are also relevant to assess water and nutrient consumption of service crops [25,26]. Indeed, root morphological traits may relate to water infiltration in soils and water acquisition [25–28]. For example, root diameter is negatively correlated with water acquisition. In contrast, root length and mass density (RLD and RD), specific root length (SRL) and the very fine root fraction (VFRf) positively correlate with water acquisition [25]. Moreover, the VFRf, SRL and root diameter are also related to the acquisitive and conservative strategies of species [29]. Finally, root traits play a role in soil stabilization processes, in natural plant communities or agrosystems [30–32]. However, the relationship between plant functional traits and ecosystem services is also complex. Single traits may be associated with multiple services and single services may be associated with multiple traits, trade-offs or synergies may exist among

different services driven by common traits or ecosystem processes [4,33], and the relationship patterns between functions and plant functional traits vary across ecosystems [34].

Multifunctionality, the ability of ecosystems to provide multiple functions and services simultaneously, is a complex concept that has been differentiated into 'ecosystem function multifunctionality' (EFM) and 'ecosystem service multifunctionality' (ESM) [35]. While EFM assesses overall ecosystem performance, ESM is defined and valued from a human perspective, making it more relevant for applied research. Despite the challenges in quantifying multifunctionality, recent advances in frameworks and metrics are improving our understanding of how different farming systems provide specific bundles of ecosystem services over time [36,37]. Biodiversity has been shown to enhance the multifunctionality of managed ecosystems, emphasizing its importance in developing management scenarios for improved ecosystem service provision [38–40]. Service crop mixtures can increase functional diversity within cropping systems. Designing mixtures with complementary plant traits may be particularly effective to increase agrosystem multifunctionality and sustainability [41]. Trade-offs between ecosystem services provided by these agrosystems depend on their composition, structure, and management [42]. For example, in a 2019 review [43], only 19% of studies explicitly identified the drivers and mechanisms underlying ecosystem service relationships. Consequently, the authors recommend that more assessments be conducted that identify the drivers of trade-offs and synergies. This can be facilitated by wider use of causal inference and process-based models, which are essential for the effective management of ecosystem services. [43].

Several agronomic challenges – such as low organic carbon levels, erosion, and fertility loss – are faced by vineyard soils, among other agrosystems. The potential of service crops to mitigate these problems has been demonstrated in numerous studies [44–46]. Beyond their contribution to enhancing soil organic matter and fertility while mitigating runoff and erosion, service crops in vineyards deliver a broad spectrum of ecosystem services [47]. These encompass the suppression of weeds, the regulation of pests and diseases, the maintenance and purification of water resources, the improvement of soil trafficability, and the conservation of soil biodiversity. However, the integration of service crops with grapevines can also have drawbacks, in particular competition for soil resources, which is often cited as a reason against their use [48].

Habitat heterogeneity and cover crop or spontaneous vegetation are crucial for biodiversity, and recent studies proved that improving biodiversity in vineyard landscapes could positively affect ecosystem services [49]. Synthetic indicators, i.e., indicators that combined several sub-indicators, are relevant to assess soil-based ecosystem services: e.g., chemical fertility for supporting plant production, physical quality for water-related services, soil morphology for erosion and flood control [50]. Garcia et al. [31,49] examined how functional markers of cover crop communities—comprising both sown and spontaneous species—relate to key viticultural functions, namely soil aggregate stability, soil water storage, and ground cover for reducing runoff and erosion. However, their work primarily focused on individual indicators in isolation and did not address multifunctionality by integrating the various indicators into a comprehensive framework.

This study aimed to identify the links between the functional structure of the service crops associated with grapevines, the function they deliver and ecosystem function multifunctionality (EFM), in a Mediterranean vineyard. Experimentally, we studied 38 plant communities each composed of one sown service crop (among 13 studied) and the spontaneous vegetation that has developed with it. At vine budburst, we simultaneously measured five functions, along with twelve functional markers of the community associated with these functions. We hypothesized that (H1) the service crops provide multiple ecosystem functions and that (H2) the mean traits of the service crops explain part of these functions. We also expected that (H3) community functional compositions affect both correlations between functions and the level of EFM on the other hand.

## Materials and methods

### Experimental site and design

The experimental vineyard was planted in 2008, and was located near Montpellier, south of France (43°31'55" N, 3°51'51" E). In 2016–2017, total rainfall over the experiment cumulated to 540 mm, with 200 mm in October 2016 only and a relatively dry spring (see Garcia et al. [30,51] for further details). Thirteen different species of service crops were chosen for

this experiment (S1 Table). In September 2016, all inter-rows were tilled for seedbed preparation. Thirteen different mono-cultures of service crop species were sown in the inter-rows (2.5 m wide) of plots of 30 m length that covered one row and the two adjacent inter-rows, at random locations. Three plots were sown for each service crop species. Species were chosen to have a diversity of botanical families (Fabaceae, Poaceae, Plantaginaceae, Boraginaceae, Rosaceae, Brassicaceae), life cycles, and growth rate, based on literature analysis (S1 Table, see also Garcia et al. [30,51]). No weeding was performed after sowing, so we obtained plant communities composed of sown and spontaneous species. Moreover, three plots were maintained with pure spontaneous vegetation. Before grapevine budburst (April 2017), three quadrats (0.25 m$^2$) were placed in each plot (total: 42 quadrats), except in 4 quadrats with Triticosecale, *Secale cereale* and *Brassica carinata* due to low service crops development, resulting in a total of 38 studied quadrats (S2 Table).

## Functional traits and markers measurements

**Aboveground vegetation sampling and trait measurements.** At grapevine budburst (April 2017), all species (sown or spontaneous) were identified in each quadrat. Then, total aboveground biomass was sampled from all quadrats (see also Garcia et al. [51]). All samples were produced within a week. For each quadrat, species were separated and weighed separately after drying (72 h, 60 °C) to record their relative aboveground biomass (precision balance PRECISA XB620C). The biomass from each quadrat was subsequently ground and sent to a laboratory to measure the carbon and nitrogen content at the community level. Aboveground traits were measured following biomass sampling in the quadrats, only on species that represented 80% of the total biomass in each quadrat [52]. For trait measurements, 15 plants per species were collected in the inter-rows of the experiment, outside the quadrat. All traits were measured according to standardized protocols [53]. Just after harvest, the plants were put in distilled water and stored for the night at 5 °C for rehydration. After rehydration, leaf fresh biomass and total plant fresh biomass were measured separately. Leaves with petioles were scanned at 400 dpi with a scanner Epson Perfection V800, and leaf area was measured using WinFOLIA software (Regent Instruments, Quebec, Canada). Then, leaves and whole plants were oven-dried at 60 °C for 72 h for dry weight determination. Specific leaf area (SLA) was calculated by dividing leaf area by dry leaf biomass, leaf dry matter content (LDMC) was calculated by dividing dry leaf biomass by fresh leaf biomass (Table 1). Dry matter content (DMC) was calculated by dividing dry plant biomass by fresh plant biomass.

**Root functional markers.** Root functional markers were measured with soil cores sampled just after biomass sampling. To improve the representativeness of the sampled area and minimize the risk of localized bias (e.g., sampling a single atypical soil point), two soil cores were collected within each quadrat and treated jointly. Each core was sectioned into five depth layers (0–10, 10–20, 20–40, 40–60, and 60–100 cm). For each layer, one half of each core was combined for root measurements, while the other halves were pooled, homogenized, and then subsampled for water content and inorganic nitrogen analysis. This approach ensured a more integrated representation of the quadrat and reduced the influence of potential spatial heterogeneity within the sampling area. All the samples were stored in a freezer at −20 °C before root measurements. After storage, the samples were thawed out in water for each soil layer. Roots were washed and separated into herbaceous roots and grapevine roots. Grapevine roots were not included in the analysis, as root absorption is low at budburst [55]. For each plant community, three subsamples of herbaceous roots were put in a clear acrylic tray with water, and roots were scanned at 600 dpi (scanner Epson Perfection V800). We used the software WinRHIZO Reg (Regent Instruments, Quebec, Canada) to analyze scanned images. As we were not able to differentiate root orders in the soil cores as suggested by McCormack et al. (2015) [56], diameter classes were used to sort out the roots: each 0.1 mm from 0 to 1 mm, each 0.5 mm from 1 to 2 mm and the class >2 mm. We included all roots in the analysis, as roots <2 mm were systematically between 98% and 100% of the total length for all plant communities.. After scanning the roots, their dry biomass was measured (precision balance SARTORIUS QUINTIX35). We estimated the total root length for each soil layer by applying the ratio of scanned root length to scanned dry biomass to the unscanned root biomass, using a proportional calculation.. Root functional markers of the community [19,57] were calculated instead of

**Table 1. List of above and belowground measured or calculated functional traits and markers, unit, ecological meaning and associated reference.**

| Functional traits or markers | Acronym | Unit | Ecological meaning | Key references |
|---|---|---|---|---|
| Ratio between carbon and nitrogen content of aerial parts | **CN** | – | Reflects the balance between structural investment (C) and metabolic activity (N); high C:N indicates conservative resource-use strategy and low decomposability. | Pérez-Harguindeguy et al. [53] |
| Depth at which the root system reaches 80% of its total length | **Depth80** | cm | Indicates rooting depth and access to deep soil resources (water, nutrients), reflecting drought tolerance and soil exploitation strategy. | Freschet et al. [54] |
| Average root diameter | **Diam** | mm | Reflects trade-offs between resource uptake efficiency (thin roots) and longevity (thick roots). | Freschet et al. [54] |
| Plant dry matter content | **DMC** | mg g$^{-1}$ | Indicates the proportion of structural material per unit fresh mass; reflects tissue toughness and conservative growth strategy. | Pérez-Harguindeguy et al. [53] |
| Leaf dry matter content | **LDMC** | mg g$^{-1}$ | Reflects leaf tissue density, longevity, and position on the leaf economics spectrum; high LDMC indicates conservative water and nutrient use. | Pérez-Harguindeguy et al. [53] |
| Plant nitrogen content | **N** | g kg$^{-1}$ | Reflects plant nutritional status and potential photosynthetic capacity; key indicator of resource acquisition strategy. | Pérez-Harguindeguy et al. [53] |
| Root length density (root length per unit soil volume) | **RLD** | cm cm$^{-3}$ | Indicates spatial exploration capacity and soil resource interception efficiency. | Freschet et al. [54] |
| Root mass density (root mass per unit soil volume) | **RD** | kg m$^{-3}$ | Quantifies belowground biomass investment; related to carbon storage and soil structure stabilization. | Freschet et al. [54] |
| Root tissue density | **RTD** | g cm$^{-3}$ | Reflects trade-off between construction cost and lifespan of roots; high RTD indicates conservative strategy. | Freschet et al. [54] |
| Specific leaf area | **SLA** | m$^2$ kg$^{-1}$ | Indicates light capture efficiency and growth rate; high SLA=acquisitive strategy, low SLA=conservative. | Pérez-Harguindeguy et al. [53] |
| Specific root length (root length per unit mass) | **SRL** | m g$^{-1}$ | Reflects efficiency of soil exploration per biomass invested; high SRL=acquisitive, low SRL=conservative. | Freschet et al. [54] |
| Proportion of very fine roots | **VFRf** | – | Indicates relative investment in absorptive roots (<0.5 mm); strongly linked to nutrient uptake dynamics. | Freschet et al. [54] |

functional traits of the species separately [12], mainly because we could not differentiate the species in soil cores, except grapevine. These markers represent the root traits of an "average plant" representing the community, as for aboveground traits [57]. We calculated root mass density (RD, kg m$^{-3}$) and root length density (RLD, cm cm-3) as total root biomass and length divided by soil volume, respectively (Table 1). The software calculated the root mean diameter (DIAM). We calculated specific root length (SRL, m g-1) as the ratio between root length and root dry mass, and very fine root fraction (VFRf) as the ratio between length of roots <0.1 mm and total root length (Table 1). Scanned samples were oven-dried (60 °C, 72h) and weighed so as non-scanned samples to extrapolate the values from the subsamples. The depth at which the communities reached 80% of the total root length (Depth80, cm) was calculated with a linear regression between cumulative total root length and soil depth using log-transformation. The root functional markers were calculated for the two soil layers 0–10 cm and 10–20 cm before aggregating them by calculating either a sum (RD, RLD) or an average (DIAM, SRL, VFRf) depending on traits. The 0–20 cm soil layer was chosen as it corresponded to the mean depth at which plant communities reached 80% of their total root length, across all plant communities (Garcia et al. 2020).

**Ecosystem functions monitoring and indicators.** We focused on five functions of major importance in viticulture: runoff reduction, soil stabilization, soil mineral nitrogen supply for the vine, soil water supply for the vine, and community biomass production. These functions were assessed at vine budburst through five indicators: the cover rate as a proxy of runoff reduction, the mean weight diameter of soil aggregates as a proxy of soil stabilization, the mineral nitrogen stock as a proxy of nitrogen supply, the soil water stock as a proxy of water supply. Finally, the biomass of the communities at the quadrat level was chosen as the indicator of aboveground production (see the previous section "Aboveground vegetation sampling and trait measurements").

 

**Runoff reduction: The cover rate.** Along with species identification during aboveground vegetation sampling, species' respective cover rate was estimated visually as a percentage of the total cover rate of the plant community, along with the total cover rate of the plant community [51]. Although runoff can also be strongly influenced by factors such as slope and soil texture, the quadrat cover rate was chosen as a proxy of potential runoff reduction (Runoff Reduction), due to its importance in reducing water runoff and improving water infiltration [58,59]. To assess the potential of runoff and erosion control of the service crops, we consider the soil to be in a favorable condition based on the minimal threshold found in the literature corresponding to a soil cover of 40% [60].

**Soil stabilization: The mean weight diameter of soil aggregates.** Soil aggregate stability was chosen as the indicator of soil stabilization (Soil Stabilization). Aggregate stability was measured using the method of Le Bissonnais [61] (ISO-10930, 2012). Only the fast-wetting disruptive test was performed here. For the 38 quadrats, the test was performed on three soil subsamples (10 g of dry soil each). Before measurements, samples were dried at 40°C for 24 h. After drying, soil aggregates were immersed in distilled water for 10 min, and transferred onto a 50 μm sieve immersed in ethanol (96°) to separate > 50 μm fragments from < 50 μm fragments. The > 50 μm fraction was collected, oven-dried (40 °C, 48 h) and sieved through a column of 6 sieves (2.00, 1.00, 0.50, 0.20, 0.10, and 0.05 mm). The mean weight diameter (MWD) was calculated to express aggregate stability, corresponding to the sum of the mass fraction of soil remaining on each sieve multiplied by the mean diameter of each sieve and its higher adjacent sieve [61]. Coarse elements (> 2 mm gravel) were weighed to correct the MWD values.

**Soil mineral nitrogen supply for the vine: The soil mineral nitrogen stock at 20 cm depth at grapevine budburst.** The soil mineral nitrogen stock in the 0−20 cm layer at grapevine budburst (April 2017) was chosen as the indicator of soil mineral nitrogen supply for the vine (Nitrogen Supply). From budburst to the 5−6 leaf phenological stage, grapevine growth depends most of all on its woody N reserves, but then a peak of N absorption starts and lasts until veraison [55]. We assumed that starting the grapevine cycle with high soil N content decreases the risk of nitrogen stress in the critical stages that follow [48]. Moreover, benefiting from a high soil N content at service crop destruction may provide a nitrogen source for bacteria that decompose plant material, depending on the C/N ratio of the service crop [62]. Consequently, at budburst, for all quadrats and the soil layers 0−10 cm and 10−20 cm, the samples originating from the two soil core halves that were not used for root measurements were sent to a laboratory for inorganic nitrogen content determination (KCl extraction through colorimetric reactions in a sequential analyzer). The nitrogen stock (Nstock, kg ha$^{-1}$) of each soil layer was calculated following the equation:

$$Nstock = NC \times D \times (100 - GC) \times BD \times 10^{-4}$$

(1)

with NC the nitrogen content (kg kg$^{-1}$), D the thickness of the soil layer (m), GC the gravel content (%, in mass) and BD the bulk density of the soil layer (kg m$^{-3}$). Total nitrogen stock was calculated by aggregating the two soil layers.

**Soil water provision: The total water stock up to 1 m depth at grapevines budburst.** The water stock in the soil up to 1 m depth was chosen as the indicator of soil water provision (Water Supply). In all quadrats and for each soil layer, the sample originated from the two soil core halves and dedicated to water measurement was weighed to determine the soil fresh mass. Then, the samples were oven dried (105 °C, 72 h), and weighed to determine their dry mass. The gravimetric soil water content (WC) was calculated. After measuring the dry mass, samples were put in a sieve (2 mm) in water to separate the soil from the gravels. Gravels were then oven-dried (105 °C, 24 h) and weighed. The water stock of each layer was calculated as follows:

$$WS = WC \times D \times (100 - GC) \times BD \, / \, Water \, density \times 10^{-4}$$

(2)

with WS the water stock (mm), WC the gravimetric soil water content (%), D the thickness of the soil layer (mm), GC the gravel content (%, in mass),BD the bulk density of the soil layer (g cm$^{-3}$), Water density (g cm$^{-3}$) and the $10^{-4}$ factor to convert ha into m$^2$.

Total soil water stock was calculated by aggregating the water stock of each soil layer between 20 cm and 100 cm depth. The 0–10 cm and 10–20 cm layers were excluded to avoid accounting for excess moisture in these topsoil layers due to recent rainfall prior to sampling. Additionally, the water stock was measured on 5 tilled inter-rows (bare soil) as controls to assess the relationship between water stock and cover rate.

We hypothesized that the water stock is well correlated with the Available Soil Water (ASW) for grapevines due to soil homogeneity in our experimental field. However, it is difficult to assess absolute water availability at this stage without information on the grapevine Total Transpirable Soil Water (TTSW). Moreover, rainfall may occur later in the season, recharge the soil water and compensate for the water deficit at budburst. For these reasons, we hypothesized that the higher the soil water content at budburst, the lower the risk of water stress would be within the grapevine cycle.

## Data analysis

All data management operations and statistical analysis were performed using R (RStudio version 2021.09.0 + 351) with packages broom, corrr, factoextra, FactoInvestigate, FactoMineR, GGally, ggcorrplot, ggpubr, ggrepel, ggsignif, gtsummary, kableExtra, multcompView, relaimpo, tidyverse.

To calculate functional leaf traits at the community scale, we assumed that the ecosystem function was determined by the more dominant species in the community (the mass-ratio hypothesis [63]). Community-weighed means (CWM [57]) were calculated for all traits following the equation:

$$CWM = \sum_{i=1}^{n} trait_i \text{ x } p_i$$

(3)

where $trait_i$ is the mean trait value of the species i, $p_i$ is the proportion in biomass of the species i in the community, and n is the number of species in the community. Throughout this article, SLA, LDMC and DMC stand for the community-weighed means of the traits in each community.

To analyze the relationships between the functional traits and markers of the service crops, we performed a principal component analysis (PCA with individuals = communities at the quadrat level, variables = community weighed means of traits, and supplementary variables = function indicators or dominant botanical family). Then we classified the quadrats into functional groups with a hierarchical clustering on principal components (HCPC). To compare the characteristics of each cluster regarding functional traits and markers and functional indicators, we performed ANOVA followed by multiple comparison tests (Tukey, p-value = 0.05).

To test the relationships between each function and the functional traits calculated at the community scale or functional markers, we first calculated Pearson correlation coefficients between the function indicator and each trait or marker. Then, we made multiple regressions between the function indicators and the coordinates of the observed communities on the first two dimensions of the PCA. Using coordinates in the PCA allowed us to reduce the number of variables, and to avoid collinearity between regressors. The relative importance of each regressor to the total variance was calculated with the function 'calc.relimp' of the package 'relaimpo' [64]. Relationships between function indicators were tested with Pearson correlations, with the ggpairs function of the package GGally.

The measured value of each function proxy was relativized to the control within the same year [65] so that a higher value always indicates greater provision. Positive values indicate that the service crop performed better than the control (hereafter, "service"). Negative values indicate that the service crop performed worse than the control (hereafter, "disservice"). For each function indicator, we subtracted the control value from each functional indicator value and then divided the result by the standard deviation to put values on a comparable scale while retaining directionality (positive = service;

negative = disservice). Multifunctionality index was calculated as the average of standardized function values for each community [66].

## Results

### Communities with different species compositions had contrasted functional descriptions

The PCA conducted on community functional markers accounted for 61.65% of the total variance across the first two axes (Fig 1A). The first axis explained 43.05% of the variance, primarily driven by VFRf and CN (positive loadings) and Diam_moy and N (negative loadings). These four variables contributed 15.7%, 14.9%, 16.2%, and 15.8% of the variation on this axis, respectively. The second axis accounted for 18.6% of the total variance, with RD and RTD as the main contributors (both with positive loadings), explaining 33.5% and 25.9% of the variation on this axis, respectively. This analysis highlights the strong discriminatory effect of root markers in differentiating plant communities.

The PCA revealed correlations among the functional markers of the community. Notably, strong positive correlations were observed between N and Diam_moy (0.72), as well as between VFRf and CN (0.7) (Fig 1A, S3 Fig). These two pairs of traits were, in turn, strongly negatively correlated with each other (between −0.67 and −0.97). Additionally, SRL and Depth80 showed a strong negative correlation (−0.54), while RD and RTD exhibited a moderate positive correlation (0.51).

The PCA plot (Fig 1B) illustrates the distribution of plant communities, each characterized by distinct species compositions (S2 Table), along the first two principal axes. The wide dispersion of communities suggests significant differences in their functional traits. The communities were also differentiated according to their dominant botanical families. Axis 1

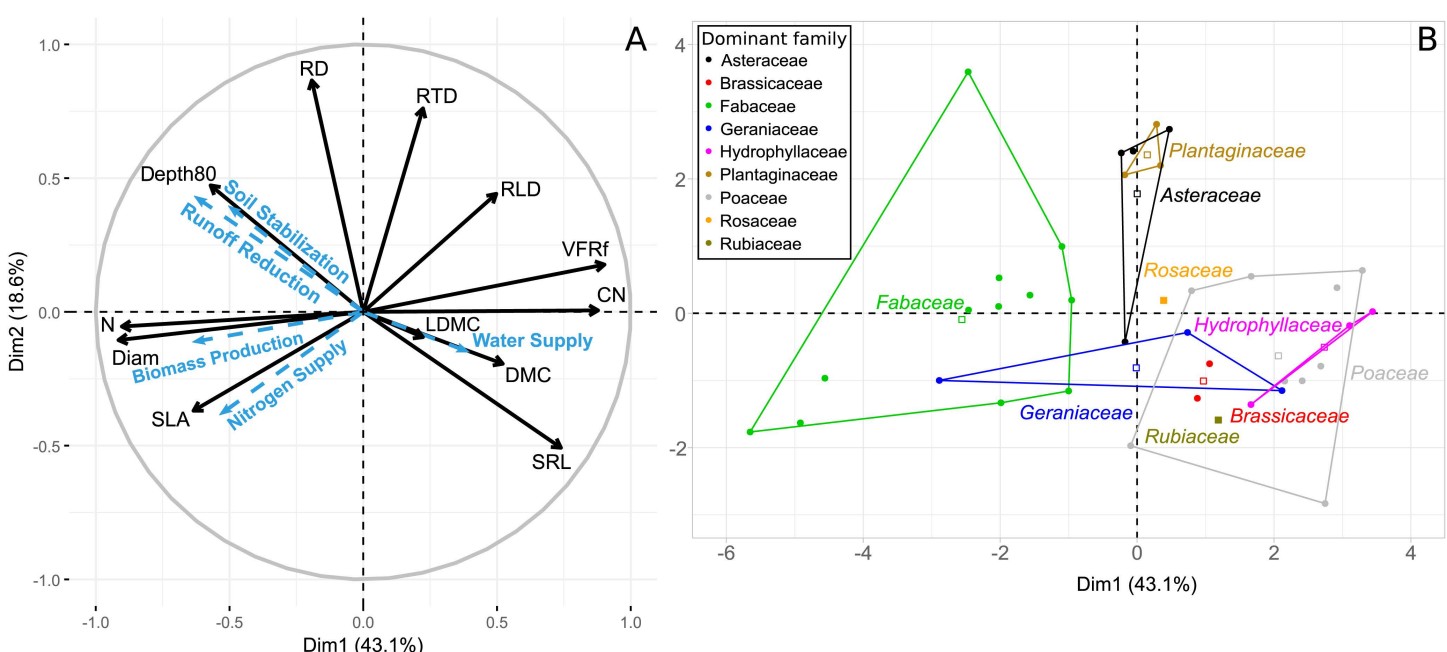

**Fig 1. Principal component analysis performed on the community functional traits and markers; A- correlation circle between variables on the first two axes (supplementary variables are represented in blue); B- graph of individuals on the first two axes, colored according to the dominant botanical family of the community.** SLA: Specific Leaf Area, DMC: Dry Matter Content, LDMC: Leaf Dry Matter Content, N: service crop nitrogen content, CN: service crop C to N ratio, VFRf: Very Fine Root Fraction, Diam: root mean diameter; SRL: specific root length; RTD: root tissue density; RLD: root length density, RD: root mass density, Depth80: the depth to reach 80% of total root length.

distinguished Fabaceae from Poaceae, Hydrophyllaceae, Brassicaceae, and Rubiaceae, whereas Axis 2 primarily separated Plantaginaceae, and to a lesser extent Asteraceae, from most other families.

## Functional markers at the community level explained part of the targeted functions

The indicators of functions were correlated with several traits or markers measured or calculated at the community level (Table 2). None of these correlations were very strong (r<0.614) and several traits or markers were significantly correlated with each function indicator (between 5 for Water Supply and 9 for Biomass Production, Table 2). Water Supply was the least correlated with functional traits and markers, exhibiting at the community scale only five moderate correlations (positive with CN, VFRf and SRL, and negative with depth80 and Diam). In contrast, Runoff Reduction and Biomass Production showed strong correlations with several functional traits or markers. Runoff Reduction was strongly positively correlated with N and Depth80 and strongly negatively correlated with LDMC and SRL. Similarly, Biomass Production was strongly positively correlated with N and Diam and strongly negatively correlated with LDMC and VFRf.

Models between the function indicators and the observation coordinates on dimensions 1 and 2 of the PCA showed that for most of the functions, the two dimensions were significant and dimension 1 remained the one explaining the larger part of the variance in the model (Table 3). Both dimensions explained between 16% and 56% of the variability of the indicators (Water Supply and Runoff Reduction respectively). Nitrogen Supply was highly negatively related to dimensions 1 and 2. Runoff Reduction and Soil Stabilization were highly negatively related to dimension 1 and positively to dimension 2. Water Supply and Biomass Production were significantly related only to dimension 1. The relation was strong and negative for Biomass Production, while low and positive for Water Supply. These results confirm the projection of function indicators observed on the PCA (Fig 1A).

## Trade-offs between ecosystem functions were observed

Across the entire dataset, only five correlations between functional indicators were significant (Fig 2). Runoff Reduction and Soil Stabilization showed the strongest positive correlation (r=0.63), followed by Biomass Production and Runoff

**Table 2. Pearson correlation coefficients between the five functions and the 12 functional traits and markers measured in the 13 plant communities.**

|  | Water Supply | Nitrogen Supply | Runoff Reduction | Biomass Production | Soil Stabilization |
|---|---|---|---|---|---|
| SLA |  | 0.421** |  | 0.406* |  |
| DMC |  |  |  | −0.374* |  |
| LDMC |  |  | −0.595*** | −0.568*** |  |
| N |  | 0.579*** | 0.549*** | 0.544*** | 0.504** |
| CN | 0.329* | −0.481** | −0.495** | −0.430** | −0.435** |
| VFRf | 0.340* | −0.521*** | −0.477** | −0.563*** | −0.363* |
| Diam | −0.332* | 0.511** | 0.501** | 0.614*** | 0.404* |
| SRL | 0.336* |  | −0.600*** |  | −0.503** |
| RTD |  | −0.404* |  | −0.367* |  |
| RLD |  | −0.414** |  |  |  |
| RD |  |  | 0.472** |  | 0.432** |
| Depth80 | −0.387* |  | 0.597*** | 0.393* | 0.461** |

Only the significant correlations are presented. ***, ** and * indicate the class of p-value associated with the correlation coefficient: ***<0.001, **<0.01, *<0.05. Positive correlations are colored in green while negative correlations are colored in red. SLA: Specific Leaf Area, DMC: Dry Matter Content, LDMC: Leaf Dry Matter Content, N: service crop nitrogen content, CN: service crop C to N ratio, VFRf: Very Fine Root Fraction, Diam: root mean diameter; SRL: specific root length; RTD: root tissue density; RLD: root length density, RD: root mass density, Depth80: the depth to reach 80% of total root length.

**Table 3. Summary of the models between the function indicators and the coordinates of observations on dimensions 1 and 2 of the PCA.**

| Function indicators | Dimension 1 | Dimension 2 | Multiple R² | p-value |
|---|---|---|---|---|
| Water Supply | 0.164 * (14.3%) | −0.095 ns (2.1%) | 0.164 | 0.044 |
| Nitrogen Supply | −0.227 *** (27.3%) | −0.247 ** (14.0%) | 0.413 | <0.001 |
| Runoff Reduction | −0.267 *** (38.0%) | 0.279 *** (17.8%) | 0.558 | <0.001 |
| Biomass Production | −0.269 *** (38.4%) | −0.072 ns (1.2%) | 0.396 | <0.001 |
| Soil Stabilization | −0.212 *** (23.9%) | 0.254 ** (14.8%) | 0.387 | <0.001 |

For each dimension, the estimate is indicated followed by ***, **, * or ns, depending on the associated p-value (***<0.001, **<0.01, *<0.05, ns>0.05). For each dimension, the relative importance to the total variance explained by the model is also indicated into brackets. Positive estimates are coloured in green while negative estimates are coloured in red.

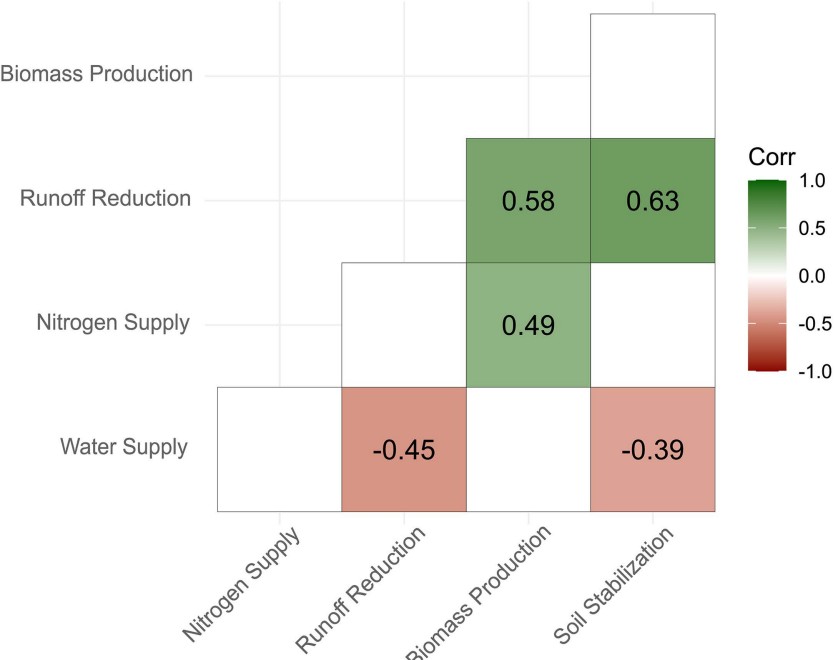

**Fig 2. Pearson correlation coefficient matrix of function indicators (n=38).** 'Corr' indicates the value of the correlation coefficient along with a color (green: positive and red: negative) gradient that indicates its strength. Only significant values (p<0.05) are presented.

Reduction (r=0.58). Nitrogen Supply was moderately correlated with Biomass Production (r=0.49) but showed no significant correlation with other indicators (|r|≤0.2, p<0.05). Finally, Water Supply was significantly negatively correlated with Runoff Reduction (r=−0.45) and, to a lesser extent, with Soil Stabilization (r=−0.39). These findings align with the correlations observed in the PCA (Fig 1).

## Functionally different communities provided different levels of functions and of EFM

The HCPC distinguished three groups of service crops (Fig 3 and Table 4). Cluster 1 and Cluster 3 differed mainly according to their positions on axis 1. Cluster 2 differed mainly from clusters 1 and 3 according to its position on axis 2.

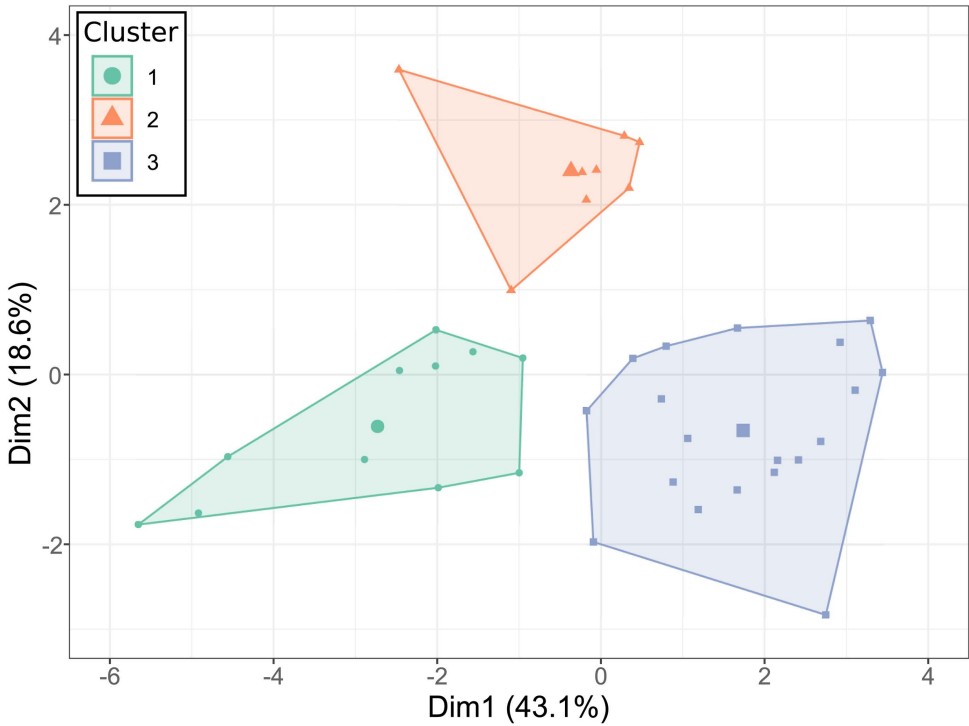

**Fig 3. Clusters of plant communities (quadrats) issued from the HCPC made on PCA coordinates of plant communities (Fig 1).**

**Table 4. Functional characterization (mean functional trait and markers and their standard deviation (sd)) of the clusters and related ANOVA p-values.**

| Functional trait or marker | cluster 1 (n = 11) | | | cluster 2 (n = 8) | | | cluster 3 (n = 19) | | | ANOVA |
|---|---|---|---|---|---|---|---|---|---|---|
| | mean | standard deviation | post hoc test | mean | standard deviation | post hoc test | mean | standard deviation | post hoc test | p-value |
| SLA (m² kg⁻¹) | **21.66** | **2.96** | **a** | 14.81 | 2.62 | b | 16.87 | 3.83 | b | 0.000194 *** |
| LDMC (mg g⁻¹) | 190.45 | 31.53 | | 178.57 | 13.91 | | 190.46 | 26.45 | | 0.526 |
| DMC (mg g⁻¹) | 170.41 | 27.85 | b | 162.52 | 17.83 | ab | 186.82 | 13.36 | a | 0.00986 ** |
| N (g kg⁻¹) | **0.90** | **0.23** | **a** | 0.40 | 0.31 | b | 0.20 | 0.20 | b | 1.61e-08 *** |
| CN | **17.55** | **2.88** | **c** | 28.83 | 7.12 | b | **35.34** | **7.20** | **a** | 5.19e-08 *** |
| VFRf | **0.53** | **0.11** | **b** | 0.71 | 0.03 | a | 0.78 | 0.07 | a | 8.77e-09 *** |
| Diam (mm) | **0.24** | **0.03** | **a** | 0.20 | 0.01 | b | 0.18 | 0.02 | b | 6.42e-07 *** |
| SRL (m g⁻¹) | 204.21 | 31.35 | b | 185.12 | 28.86 | b | **334.89** | **49.20** | **a** | 1.77e-11 *** |
| RLD (cm cm⁻³) | **22.59** | **8.65** | **b** | 33.44 | 9.98 | a | 32.12 | 11.84 | a | 0.0434 * |
| RTD (g cm⁻³) | 0.12 | 0.02 | a | 0.25 | 0.15 | a | 0.15 | 0.02 | b | 5.44e-05 |
| RD (kg m⁻³) | 0.29 | 0.27 | b | **0.77** | **0.18** | **a** | 0.11 | 0.35 | b | 3.57e-05 *** |
| Depth80 (cm) | 20.72 | 7.89 | a | 25.97 | 7.80 | a | **11.93** | **4.42** | **b** | 1.2e-05 *** |
| Dominant botanical families | Fabaceae (+) 90.9% | | | Plantaginaceae (+) 37.5% | | | Poaceae (+) 47.4% | | | |
| | Poaceae (-) 0% | | | Asteraceae (+) 37.5% | | | Fabaceae (-) 0% | | | |

Different letters indicate significant differences between clusters (post-hoc Tukey test, p<0.05). Text in bold indicates the more striking characteristics of the clusters (i.e., characteristics for which the value of the cluster is significantly higher or lower from all the other two clusters). For botanical families, the test (v-test) compares the cluster values to the overall values. % indicates the percentage of communities of the cluster for which this family was dominant. (+) (respectively (-)) indicates that this percentage is significantly higher (respectively lower) than the percentage of communities of the whole dataset for which this family was dominant.

Cluster 1 was characterized by high SLA, N, and Diam, and low CN, VFRf, RLD, and DMC—significantly higher and lower, respectively, than for the other clusters, except DMC, which was not significantly different from Cluster 2. The dominant family in Cluster 1 communities was primarily Fabaceae (91% of communities), with no community having Poaceae as the dominant family. Cluster 3 exhibited high CN, SRL, and DMC, and low Depth80—significantly higher and lower, respectively, than in the other clusters, except for DMC, which was not significantly different from Cluster 2. The dominant family in Cluster 3 communities was Poaceae (47% of communities), and no community had Fabaceae as the dominant family. Cluster 2 was uniquely characterized by significantly higher RD compared to the other clusters. The dominant families in Cluster 2 communities were Plantaginaceae and Asteraceae, each representing 37% of the communities.

Fig 4 represents the functional characterization of the clusters through the characterization of 5 ecosystem functions. Highest positive functions indicators were found for Runoff Reduction (2.14 to 3.52). For all clusters, Soil Stabilization, Runoff Reduction and Biomass Production were found positive while Water Supply was found negative. In average, nitrogen Supply was negative for clusters 2 and 3, and slightly positive for cluster 1. Water Supply and Soil Stabilization were not found to be significantly different between clusters. Interestingly, we found significant differences for the other function indicators. In contrast, Nitrogen Supply was identified as significant when comparing clusters 2−1 and clusters 3−1.

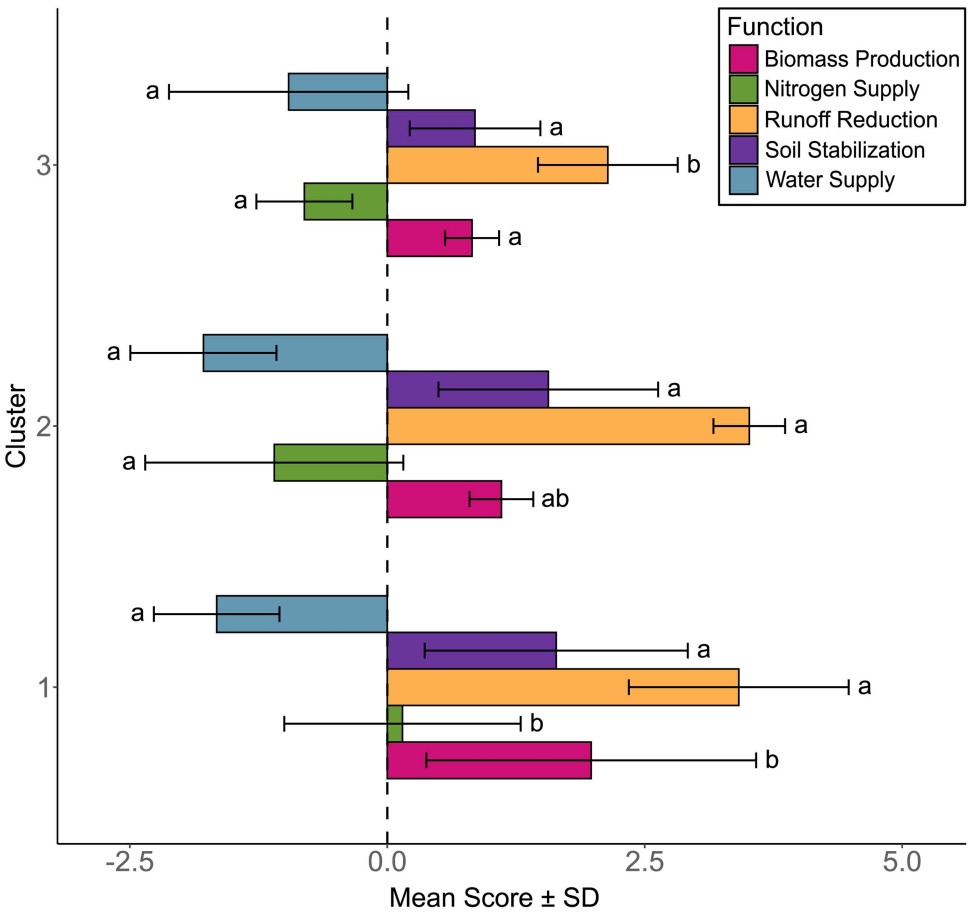

**Fig 4. Comparison of service crop clusters with respect to the five functions.** The mean scores are presented relative to bare soil, with the standard deviation shown for each function. For each function, different letters indicate significant differences between clusters (Tukey test, $p < 0.05$).

Likewise, Runoff Reduction was observed to be statistically significant when comparing clusters 3−1 and clusters 3−2. Lastly, Biomass Production demonstrated statistical significance only between clusters 3−1.

The Multifunctionality index exhibits significant variations across clusters, with a mean value of 1.1 for Cluster 1, 0.66 for Cluster 2, and 0,41 for Cluster 3 (Fig 5) and a significant difference between Cluster 1 and Cluster 3 (Tukey test, p = 0.0007559).

## Discussion

We explored the relationships between plant communities in Mediterranean vineyard inter-rows and a set of soil and plant indicators, considered as proxy of ecosystem functions. Our results suggest that temporary service crops in Mediterranean vineyards based on spontaneous vegetation increase soil-based ecosystem functions. We examined how certain plant traits of these plant communities are connected to these soil-based functions, revealing that there are trade-offs between these functions.

### The service crops provide multiple ecosystem functions

Overall, in accordance with our first hypothesis, the three functional groups of spontaneous vegetation communities we studied in a Mediterranean vineyard have a positive multifunctionality index (Fig 5) in comparison with bare soil. More precisely, independently from their functional characterization, the groups of communities simultaneously provided three functions: Soil Stabilization, Runoff Reduction and Biomass Production (Fig 4). These results are consistent with the literature.

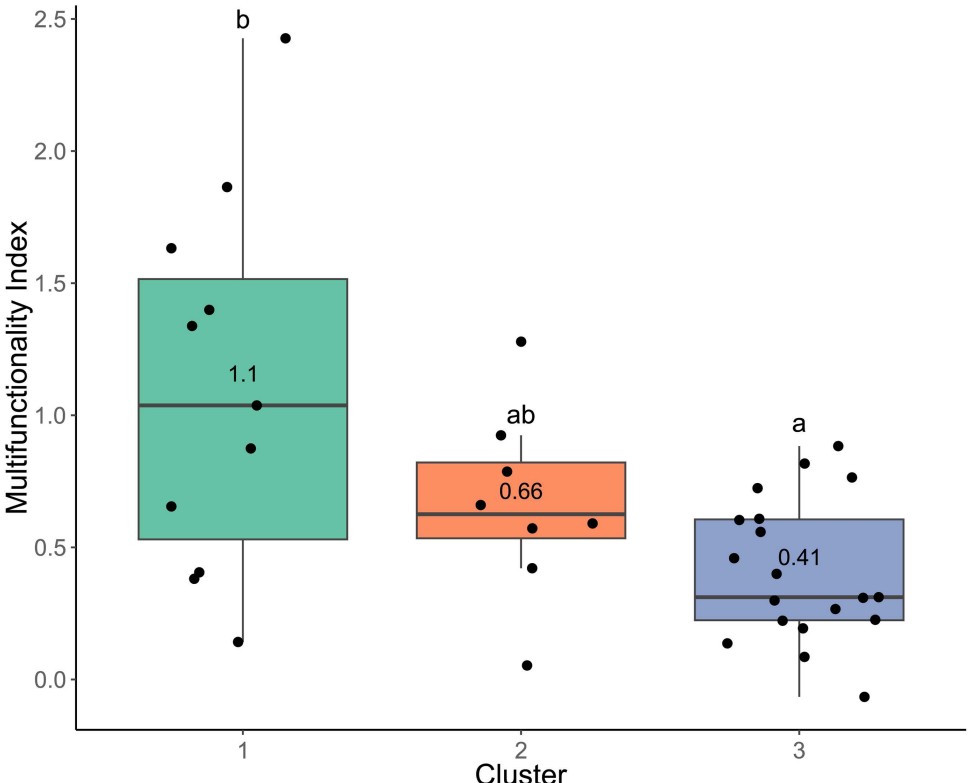

**Fig 5. Multifunctionality index comparison between the clusters.** Each point represents a community. Cluster mean values are indicated. Different letters indicate significant differences between clusters (Tukey test, p < 0.05).

Indeed, service crops (SCs) have been proved to provide a protective layer to the soil that significantly reduces soil runoff and erosion by minimizing both raindrop impact and the amount of exposed soil surface [67]. Although we did not directly measure runoff, the cover rate used as a proxy showed coherent results regarding to the processes involved in runoff reduction. Additionally, SCs enhance soil structure by increasing organic matter and promoting aggregate stability, which both contribute to maintaining soil quality over time [68]. Furthermore, the management of SCs has been shown to reduce annual runoff coefficients [69], particularly in regions with higher rainfall. Biomass production was significantly increased compared to bare soil management through soil tillage but varied considerably due to species composition (Fig 4). From a wider perspective, biomass production variability also depends on factors such as climatic conditions and management practices. In a synthesis of data from 389 published studies, Ruis et al. [70] analyzed biomass production across 20 common SCs species in temperate regions: according to this review, in drier regions receiving less than 750 mm of annual precipitation, biomass production varied greatly (ranges from 0.87 to 6.03 Mg ha$^{-1}$ of dry matter). Importantly, we found that biomass plays a key role in influencing multifunctionality (Fig 2). However, even if runoff reduction and soil stability are supposed to increase water infiltration [71], in our study, the provision of the two other functions (Water Supply and Nitrogen Supply) is either negligible compared to bare soil or negative in the case of water and is more dependent on the community. This aligns with studies highlighting competition between cover crops and the associated perennial crop (Fig 4, e.g., Durán Zuazo and Rodríguez Pleguezuelo [58]).

In this study, we employed easy-to-measure field indicators as proxies for the intricate and dynamic ecosystem functions and used a mean value to assess the global multifunctionality of different service crop communities, in comparison with bare soil, as suggested by Finney et al. [66]. We can discuss the suitability of using such simplified indicators. For example, Runoff Reduction was not directly estimated by water fluxes field measurements, but estimated from vegetation cover rate and was analyzed considering a commonly used threshold [51,60] without considering rainfall intensity patterns [72], or root traits influencing water infiltration such as root density [73]. While such indicators are interesting because they are easy to implement, they only provide a rough understanding of the mechanisms and should only be used to provide a general, comparative evaluation of vegetation cover. Similarly, while multifunctionality scores are useful to synthesize the interest of introducing service crops, however they are integrative and may hide some interesting trade-off between functions, particularly those about soil management which is a major issue for vineyard sustainability [74].

Overall, such bundles of functions have already been observed in other studies on cover crops, both for cover crop monoculture and mixtures (e.g., Finney et al. [66]). If we consider the combinations of soil functions considered, our study highlighted a major trade-off between Runoff Reduction, Soil Stabilization and Water Supply, which is consistent with previous studies in the same context [30,51] or others [75–77]. Contrary to Aryal et al. [78], we did not find that biomass production posed the highest tradeoffs, we found significant and positive correlations between Biomass Production and Runoff Reduction and Nitrogen Supply (Fig 2, 0.58 and 0.49 respectively) but negative with Water Supply. This trade-off raises major questions about SCs design and management [79], particularly in the Mediterranean region. A recent review emphasized that the full potential of SCs remains underexploited due to several knowledge gaps that need to be addressed in the short term [47]. Among the key issues identified are questions related to the characterization of trade-offs between ecosystem services: does optimal management of SCs provide a trade-off solution between provisioning, supporting and regulating services? Can management strategies reduce the ecosystem disservices associated with SCs?

## Functional traits and markers of the service crops explain part of these functions

In our study, the combination of functional markers, reflected by the two dimensions of the PCA, explained in the better case half of the observed variability in function indicators. This result was observed to a lesser degree by Garcia et al. [30] for vineyards, van der Plas et al. [15] for grassland species, and Wei et al. [80] for tree systems but the magnitude of the correlation between traits and functions has been the subject of recent debates [13,30,81–84].

First, these correlations greatly depend on functions [13,81]. In our study, functions related to the aboveground indicators of the agrosystem (Biomass Production, Runoff Reduction) were strongly correlated to more traits than functions related to the belowground indicators (Nitrogen Supply, Water Supply, Soil Stabilization). These functions operate at different levels of integration, which may explain these results. Biomass production, for example, is an integrative function that results from multiple underlying processes at the plant level, such as photosynthesis, nutrient uptake, and carbon allocation, each of which may be associated with different functional traits.

Several factors have been identified in the literature to explain the relatively low correlations between traits and functions sometimes observed in the literature. These include the selection of traits—whether they are widely used versus function-specific, or easy-to-measure versus more complex ('hard') traits [81–83], —as well as variations in the microenvironment, such as microclimate or microfauna dynamics [13,84]. Additionally, the timing of community measurements in relation to ecosystem functions can also influence these correlations [82]. In our study, while functional markers were chosen on a mechanistic basis, our analysis strategy using combinations of all markers measured (and not only the ones likely related to the target function), issued from the PCA, may have lessened the correlations with functions. We have also chosen to use simple morphological traits, known as soft traits [85], for their ease of measurement in large-scale experiments. However, soft traits are worse predictors of functions than hard traits [86]. For example, the relationships between SRL, root mean diameter, VFRf and MWD we observed may be related to water-soluble compound concentration in root tissues and the subsequent interactions with microbial biomass and fungi [29,30,87]. Indeed, root water-soluble compound concentration is negatively related to SRL and positively related to root diameter: directly measuring root water-soluble compound concentration would probably have improved the relationship between this trait and MWD. Concerning variations in the microenvironment, although our work was conducted on a rather homogeneous field, we could not exclude small differences in the microenvironment (e.g., porosity, nutrient levels, microfauna) due to local topography or cultural practices history. Finally, we assumed that the ecosystem functions we studied responded to the mass-ratio hypothesis [63], i.e., they are determined by the more dominant species in the community, and then are best related to the mean trait value of the community (CWM). However, the other moments of the trait-abundance relationship (variance, skewness, kurtosis), may improve the links between traits and functions [82].

Additionally, the limited explanatory power of traits for the studied functions can be interpreted within the framework of effect and response traits [14]. In this study, functional traits were considered as effect traits; hovever, the observed relationships also highlight varying community responses to environmental conditions. For example, VFRf and SRL were positively correlated with Water Supply, even though these acquisitive traits should have led to water depletion in the soil. However, these traits were primarily exhibited by communities dominated by grass species, which performed poorly in this vineyard soil and produced little biomass. Consequently, these communities provided limited soil cover and consumed little soil water. Here, SRL and VFRf are only weakly linked to water provisioning functions, and in a counterintuitive manner, suggesting they should be interpreted as response traits rather than effect traits.

As observed in other studies, we showed that several traits are related to the same function and that a trait can be related to several functions, leading to potential trade-offs between functions (see 4.1). These relationships are summarized in Fig 6. Some of these highlighted traits-function may be discussed here. For example, plant aboveground N content and SLA were positively related with Nitrogen Supply. This result seems consistent as these traits were mainly exhibited by leguminous species. Previous studies show that rhizodeposition exists with legume root systems [88–90]. Fustec et al. [91] showed that legumes can release from 10% to 57% of their total plant N by their roots. Moreover, Ofosu-Budu et al. [87] found a positive correlation between root size and rhizodeposition, which could support our results of a positive relationship between Diam and Nitrogen Supply. We also found that LDMC and plant aboveground N content were, respectively negatively and positively, related to Runoff Reduction and Biomass Production. These results are consistent with the fast-slow leaf economic spectrum [24] as high plant N content and low LDMC depict acquisitive strategies of plants.

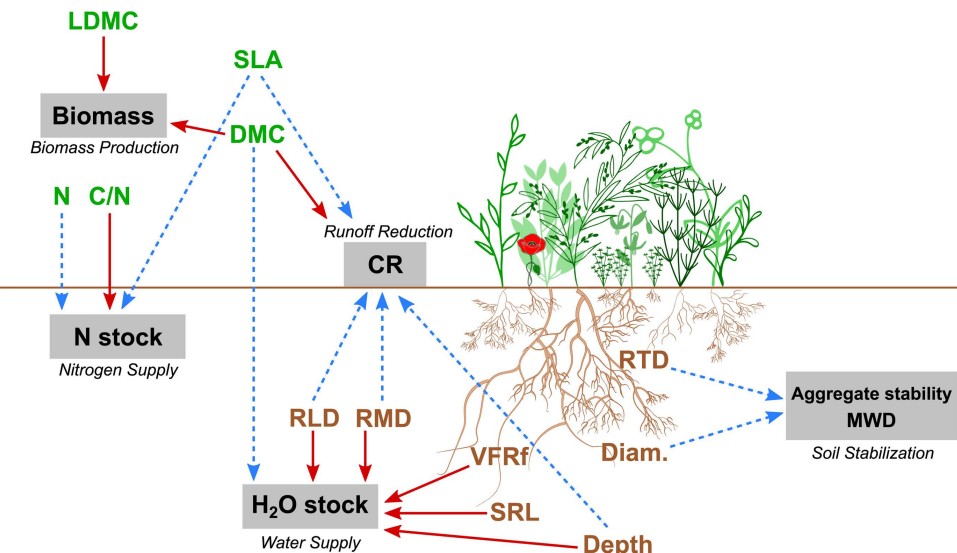

**Fig 6. Relationships between service crop functional traits and markers (green for aboveground traits, brown for belowground traits) and ecosystem function indicators (grey boxes) shown in this study.** Blue dashed arrows indicate positive relationships, red arrows indicate negative relationships. CN: plant aboveground carbon to nitrogen ratio; CR: cover rate; Depth: rooting depth; Diam: average root diameter; DMC: plant dry matter content; LDMC: leaf dry matter content; MWD: mean weight diameter; N: plant aboveground nitrogen content; RLD: root length density; RD: root mass density; RTD: root tissue density; SLA: specific leaf area; SRL: specific root length; VFRf: very fine root fraction.

## Design and management of service crop strategies to increase agrosystem multifunctionality

Enhancing multifunctionality in agrosystems requires a diversity of species within the cover crop mixtures [65,92]. Species diversity supports resource capture and promotes ecosystem services, but its relationship with biomass production is not always straightforward. While higher species richness can sometimes lead to increased biomass [65,93], monocultures may outperform mixtures in certain cases [94]. However, it is possible to achieve both high diversity and high biomass simultaneously, which should be the primary objective when designing mixtures to maximize ecosystem functions. Indeed, mixtures that include species with complementary traits can exhibit higher species evenness, multifunctionality and productivity [95,96], and contribute to the overall sustainability of the agrosystem [41].

Our study confirmed that the functional approach is a valuable tool to assess the roles of plant species and to select those that maximize multifunctionality (Fig 5). By leveraging functional traits, species can be chosen not only for their individual contributions but also for their complementarity within mixtures. This allows for an optimized design of species proportions to balance trade-offs and enhance the delivery of multiple ecosystem services [95]. For instance, in vineyard soils with low mineral content, introducing legumes into cover crop mixtures can favor biomass production, supply nitrogen to the soil and reduce competition for this resource with the grapevine [97]. Our study also illustrates the potential of such an approach: Cluster 1 outperformed Cluster 3 in terms of nitrogen provisioning while maintaining an intermediate score for soil water consumption (Fig 5). However, none of the clusters displayed positive scores for the water supply function, indicating challenges in addressing water availability alongside other services.

Beyond species selection, management practices play a crucial role in maximizing multifunctionality and balancing trade-offs among ecosystem services. For example, managing sowing and termination dates can significantly influence biomass production and the provision of services. Garcia et al. [94] showed that allowing vegetation to grow until budburst can lead to a two- to threefold increase in biomass compared to early termination. Regarding water competition with grapevines, earlier termination of the service crop or reducing its spatial coverage within the field are efficient strategies

to mitigate competition [98–100]. In some situations where this practice is allowed, irrigation can also be a key lever, as studies have shown that cover crops do not necessarily exacerbate water competition compared to weed control in irrigated systems [101,102]. Similarly, advancing the termination date can improve residue mineralization, facilitating nitrogen availability, which could also be supplemented through targeted fertilization practices post-termination.

Overall, these findings provide operational insights for the design of service crop strategies that support ecosystem services while maintaining crop performance. Integrating such functional and management-based approaches into decision-making frameworks—such as multi-criteria assessment tools, participatory modeling and co-design approaches [103], or sustainability assessment frameworks used in land-use planning (e.g., Therond et al., [104])—could help guide sustainable land-use management. Embedding the functional trait approach within these frameworks [105] would strengthen the capacity of farmers, advisors, and policymakers to jointly evaluate trade-offs, prioritize ecosystem services, and design more resilient and multifunctional viticultural landscapes.

## Conclusion

In this study, we characterized 38 spontaneous vegetation communities in a Mediterranean vineyard developed from both sown and spontaneous species, to test our hypotheses. We analyzed the relationships between the functional traits of these communities and soil-based ecosystem functions, particularly in terms of water and nitrogen fluxes and soil conservation. Consistent with H1, all communities provided multiple soil-based ecosystem functions compared with bare soil, highlighting the general benefit of soil cover. Supporting H2, biomass production—a key community-mean trait—was determinant for both agricultural and environmental benefits, particularly in relation to water, nitrogen, and soil conservation functions. In line with H3, community functional composition influenced correlations among functions and the level of ecosystem multifunctionality. Enhancing biomass production through species mixtures or adapted management (sowing and termination dates, tillage) thus represents a promising avenue for agroecological viticulture. Further validation of this functional approach across diverse cropping systems and pedoclimatic conditions could help identify ideotypes or mixtures of service crops tailored to deliver multiple ecosystem services.

## Supporting information

**S1 Table. Sown species chosen for the experimentation and their respective botanical family.**
(PDF)

**S2 Table. Composition of each plant community, relative biomass of each species (%) and total biomass of the community (t ha$^{-1}$).**
(PDF)

**S3 Fig. Pearson correlation coefficient matrix of functional markers of the communities (n = 38).**
(PDF)

## Acknowledgments

We express our gratitude to Clément Énard, Yvan Bouisson, and Bénédicte Ohl for their invaluable work in the experimental vineyard, from grapevine management to data sampling. We also thank Aurore Martenot for her contributions, from plant sampling and aboveground trait measurements to the initial stages of explanatory data analysis. Special thanks to Jean-Luc Belotti for his expertise and advice on aggregate stability measurements, and to Catherine Roumet, Florian Fort, and Christophe Jourdan for sharing their experience in root analysis and providing valuable guidance. We are particularly grateful to Inti Ganganelli, Nicolas Fleureau, and Élise Rivière for their essential assistance in root trait measurements, aggregate stability assessments, and explanatory data analysis. Additionally, we extend our appreciation to Guillaume

Fried and Jean Richarte for their help with species identification in the experimental field. Finally, we thank the reviewers that helped us to improve this article throughout the revision process.

## Author contributions

**Conceptualization:** Léo Garcia, Aurélie Metay, Gaëlle Damour.

**Data curation:** Léo Garcia, Aurélie Metay, Gaëlle Damour.

**Formal analysis:** Léo Garcia, Aurélie Metay, Gaëlle Damour.

**Funding acquisition:** Aurélie Metay.

**Investigation:** Léo Garcia.

**Methodology:** Léo Garcia, Aurélie Metay, Gaëlle Damour.

**Supervision:** Aurélie Metay, Gaëlle Damour.

**Validation:** Léo Garcia, Aurélie Metay, Gaëlle Damour.

**Visualization:** Léo Garcia, Aurélie Metay, Gaëlle Damour.

**Writing – original draft:** Léo Garcia, Aurélie Metay, Gaëlle Damour.

**Writing – review & editing:** Léo Garcia, Aurélie Metay, Gaëlle Damour.

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
