## [Decision Letter · Decision Letter 0]

1 Aug 2025

PONE-D-25-16076
Assessing the Multifunctionality of Service Crops in Mediterranean Vineyards Using a Functional Trait Approach
PLOS ONE

Dear Dr. Damour,

Thank you for submitting your manuscript to PLOS ONE. After careful consideration, we feel that it has merit but does not fully meet PLOS ONE’s publication criteria as it currently stands. Therefore, we invite you to submit a revised version of the manuscript that addresses the points raised during the review process.

We look forward to receiving your revised manuscript.

Kind regards,

Raed Abduljabbar Haleem, Ph.D

Academic Editor

PLOS ONE

https://linkinghub.elsevier.com/retrieve/pii/S1360138522000863

https://hal.science/hal-01614417v2/document

https://www.frontiersin.org/journals/sustainable-food-systems/articles/10.3389/fsufs.2020.564197/full

In your revision ensure you cite all your sources (including your own works), and quote or rephrase any duplicated text outside the methods section. Further consideration is dependent on these concerns being addressed.

“LG, AM: FertilCrop project, FP7 ERA-Net program CORE Organic Plus”

“This research was supported by financial funding for research activities conducted within the FertilCrop project, under the FP7 ERA-Net program CORE Organic Plus. We express our gratitude to Clément Énard, Yvan Bouisson, and Bénédicte Ohl for their invaluable work in the experimental vineyard, from grapevine management to data sampling. We also thank Aurore Martenot for her contributions, from plant sampling and aboveground trait measurements to the initial stages of explanatory data analysis. Special thanks to Jean-Luc Belotti for his expertise and advice on aggregate stability measurements, and to Catherine Roumet, Florian Fort, and Christophe Jourdan for sharing their experience in root analysis and providing valuable guidance. We are particularly grateful to Inti Ganganelli, Nicolas Fleureau, and Élise Rivière for their essential assistance in root trait measurements, aggregate stability assessments, and explanatory data analysis. Additionally, we extend our appreciation to Guillaume Fried and Jean Richarte for their help with species identification in the experimental field.”

“LG, AM: FertilCrop project, FP7 ERA-Net program CORE Organic Plus”

Reviewers' comments:

Reviewer's Responses to Questions

**Comments to the Author**

1. Is the manuscript technically sound, and do the data support the conclusions?

Reviewer #1: Yes

Reviewer #2: Yes

2. Has the statistical analysis been performed appropriately and rigorously?

Reviewer #1: Yes

Reviewer #2: Yes

3. Have the authors made all data underlying the findings in their manuscript fully available?

Reviewer #1: No

Reviewer #2: Yes

4. Is the manuscript presented in an intelligible fashion and written in standard English?

Reviewer #1: Yes

Reviewer #2: Yes

5. Review Comments to the Author

Reviewer #1: Dear authors,

I have carefully read your manuscript entitled “Assessing the Multifunctionality of Service Crops in Mediterranean”, and I would like to sincerely congratulate you on such a well-executed and valuable piece of research. Overall, I found the manuscript to be clearly written, methodologically solid, and rich in insights that contribute meaningfully to the field of plant functional ecology in anthropogenic contexts. However, I would like to suggest a few points that, in my opinion, could further strengthen the work prior to publication:

1. I believe the manuscript would benefit from a deeper discussion in the introduction and discussion sections regarding the “afterlife effects” of leaf functional traits and their influence on multiple litter functions. This would be particularly relevant for interpreting your results related to nitrogen fluxes and soil conservation. In this regard, I strongly recommend considering the following reference:

Dias, A.T.C., Cornelissen, J.H.C. & Berg, M.P. (2017). Litter for life: assessing the multifunctional legacy of plant traits. Journal of Ecology, 105(5), 1163–1168. https://doi.org/10.1111/1365-2745.12763

2. I would recommend re-evaluating the use of the term “runoff reduction” as one of the assessed functions. Since actual runoff was not measured directly in your study, and rather inferred from total cover rate of plant community, it would be more accurate to describe this as a proxy. As runoff can also be strongly influenced by factors such as slope and soil texture, I suggest acknowledging these limitations more explicitly in the text.

3. Although the manuscript includes nearly a hundred references, a few of them could be better aligned with specific statements. I suggest incorporating the following references, which may provide more precise support:

o Lines 55–56:

Castillo-Figueroa, D., Soler-Marín, D., & Posada, J.M. (2025). Functional Traits and Species Identity Drive Decomposition Along a Successional Gradient in Upper Andean Tropical Forests. Biotropica, 57: e13425. https://doi.org/10.1111/btp.13425

o Line 413:

Álvarez-Rogel, J., Peñalver-Alcalá, A., & González-Alcaraz, M.N. (2022). Spontaneous vegetation colonizing abandoned metal(loid) mine tailings consistently modulates climatic, chemical and biological soil conditions throughout seasons. Science of the Total Environment, 838, 155945. https://doi.org/10.1016/j.scitotenv.2022.155945

o Lines 489–490:

Castillo-Figueroa, D., González-Melo, A., & Posada, J.M. (2023). Wood density is related to aboveground biomass and productivity along a successional gradient in upper Andean tropical forests. Frontiers in Plant Science, 14, 1276424. https://doi.org/10.3389/fpls.2023.1276424

o Line 492:

Lavorel, S., & Garnier, E. (2002). Predicting changes in community composition and ecosystem functioning from plant traits: revisiting the Holy Grail. Functional Ecology, 16, 545–556. https://doi.org/10.1046/j.1365-2435.2002.00664.x

o Lines 532–533:

Kümmerer, R., Heuermann, D., Laidig, F., Piepho, H.-P., & Bauer, B. (2025). Strategies to develop simple multi-species cover crop mixtures to enhance aboveground biomass and quality. Field Crops Research, 333, 110014. https://doi.org/10.1016/j.fcr.2025.110014

4. Finally, I think the manuscript would gain additional relevance if the main findings were briefly placed within a broader land-use management context. Although I understand that policy application may not be the central focus of the study, one or two sentences connecting your findings to decision-making processes or practical implications for sustainable agriculture would add value and accessibility for broader audiences.

Once these aspects are addressed, I believe your manuscript will represent a highly valuable contribution to the fields of functional ecology and sustainable land-use management.

Reviewer #2: I hope these suggestions will improve the structure of your article and by acting on them, you will increase the speed of conveying the content to the readers.

Title:

Adjust the term ‘crops’, since throughout the document emphasis has been placed on agrosystems, ecosystems, vineyard landscapes, and plant communities. Could be: agrosystems with grapevines.

ABSTRACT

Line 12-20: It is suggested to focus on the problem and justification. Summarize. Give more relevance to your findings.

It is suggested to include the treatments and experimental design. Likewise, mention which variables were measured. Statistical analysis too.

Line 27: Correctly denote the regression coefficient.

Line 32: How was sustainability measured in the agrosystem?

INTRODUCTION

Line 37-67: It is suggested that some numerical data be added to strengthen the variables in question. For example, how much productivity increases under these diversified systems.

Line 94: What does ES mean?

Line 94-95: Clarify what is a synthetic indicator?

Line 97: In what phenological states of the grapevine plant were these parameters measured?

It is suggested to include the economic importance of this type of vineyard system and the proportion in which it exists in the country. Also, in terms of public policies, what has been done for the management of these systems?

It is suggested to include the knowledge gap regarding the variables and the research subject.

Material and methods

It is suggested to add the name of the equipment and instruments that were used to measure the variables in question (brand, model, condition and country of manufacture).

It is suggested to add the type of climate and soil at the experimental site.

It is suggested to add a representative Figure of how the treatments and replicas were distributed.

Explain why these 13 plant species were used for the experiment. Maybe the answer is on line 120, but there's no connection between the paragraphs. Clarify this point.

Based on what method were the 13 botanical families used in the experiment determined?

Also specify whether the experiment was established in the open field or under greenhouse conditions.

Line 124: Were the quadrants established permanently or temporarily?

Line 129: It is suggested to add the age of the plant.

Line 153: Specify the number of subsamples of herbaceous roots.

Line 172: Please specify why it was not calculated for all 5 soil layers?

In Table 1, we suggest adding the authors from which these variables were taken. Also, in the first column, describe the acronym in full and include the acronym in parentheses.

Line 180: add at what age of the plants these variables were taken.

Line 198: What is the degree of purity of ethanol?

Line 222: Justify why up to 1 m deep? and not 2 meters. Support with a bibliographic citation.

Line 248-249: For what subsequent statistical analysis were these weighted means used?

Line 257-258: Why was a dendrogram not made with Euclidean or Gower distances?

Results

Line 280: It is suggested to add how many axes add up to 100% of the total variance.

Line 282-283: The sum of the contributions of each variable is 62.6%. Verify.

Line 290-291: What does moderate positive correlation mean? It's better to give values.

Line 292: Figure 2A is not in the document. Please verify.

In Figure 1 separate the words from the supplementary variables.

In Figure 1 add Quadrants I, II, III and IV.

Line 295-297: Specify how the botanical families were distinguished in both axes 1 and 2, since this was to be expected since they were different species.

Line 299: Add the subject of study.

Line 318: Replace ‘experiment’ with ‘13 plant communities’. Please verify.

Line 371: Add the subject of study.

Line 373-374: what does resp. mean?

In Figure 4 also separate the words from the variables

In Figure 3B, what do the orange triangles, green circles, and blue squares represent? Label what they correspond to.

In Figure 6, in the plants, represent the grapevine crop.

Discussion

In this section, repeat the same subheadings from the Results to improve the study’s flow. It’s not enough to simply state that the results are the same or different from those of other researchers; you must explain why. Please also add the limitations and future work arising from the results of the present study.

Why was an economic analysis not included to determine which treatment would be most appropriate to recommend to grapevine growers?

Conclusion

They must be concise and based on the hypotheses raised.

6. PLOS authors have the option to publish the peer review history of their article (what does this mean?). If published, this will include your full peer review and any attached files.

Reviewer #1: No

Reviewer #2: No

---

## [Author Response · Author response to Decision Letter 1]

17 Oct 2025

All the editorial and reviewer’s comments were considered in this revised manuscript. A point by point responses is provided in the "Response to Reviewers_R1" file.

---

## [Decision Letter · Decision Letter 1]

13 Jan 2026

PONE-D-25-16076R1
Assessing the Multifunctionality of Service Crops in Mediterranean Vineyards Using a Functional Trait Approach
PLOS One

Dear Dr. Damour,

Thank you for submitting your manuscript to PLOS ONE. After careful consideration, we feel that it has merit but does not fully meet PLOS ONE’s publication criteria as it currently stands. Therefore, we invite you to submit a revised version of the manuscript that addresses the points raised during the review process.

We look forward to receiving your revised manuscript.

Kind regards,

Raed Abduljabbar Haleem, Ph.D

Academic Editor

PLOS One

Journal Requirements:

Reviewers' comments:

Reviewer's Responses to Questions

**Comments to the Author**

1. If the authors have adequately addressed your comments raised in a previous round of review and you feel that this manuscript is now acceptable for publication, you may indicate that here to bypass the “Comments to the Author” section, enter your conflict of interest statement in the “Confidential to Editor” section, and submit your "Accept" recommendation.

Reviewer #3: (No Response)

Reviewer #4: (No Response)

2. Is the manuscript technically sound, and do the data support the conclusions?

Reviewer #3: Yes

Reviewer #4: Yes

3. Has the statistical analysis been performed appropriately and rigorously?

Reviewer #3: Yes

Reviewer #4: Yes

4. Have the authors made all data underlying the findings in their manuscript fully available?

Reviewer #3: Yes

Reviewer #4: Yes

5. Is the manuscript presented in an intelligible fashion and written in standard English?

Reviewer #3: Yes

Reviewer #4: Yes

6. Review Comments to the Author

Reviewer #3: This article assesses the contribution to soil and agronomic traits of a number of different service crop communities in a French vineyard. The research is described adequately and analysis is in my opinion sufficiently rigorous. However, there are some shortcomings in some of the reporting, leading to parts of the paper which are hard to understand. There are cases where there could have been more attention to detail in manuscript preparation (eg figure numbering).

I note that this paper has been reviewed before and I am happy that comments from the two previous reviewers have for the most part been addressed.

--

Data are available and can be downloaded from the link provided. Please ensure filenames match filenames given in ReadMe.tab (csv not tab) and files have proper separators (eg you list a .tab file but separators are semicolon).

Abstract L41 I don't understand the wording 'determinant to provide soil-based ecosystem functions'

L51 could you give these as % increases, either as well as or instead of the mass per hectare: it is unclear whether these yield increases are of significance.

L95 I'm not sure six years ago counts as 'recent' any more. Maybe state '2019' instead of 'recent'.

L132/3 I'm not sure this is an 'on the one hand' situation - you could just say 'both correlations between functions and the level of EFM'.

Reviewer 1 Comment 1: You have not included a reference to this paper; have you considered it? I can't find any mention on line 80 (original or revised) to leaf functional traits in relation to litter decomposition - possibly L65 (original) or L72 (revised). It feels like this comment has not been properly addressed.

Reviewer 1 Comment 2: It still looks like you measured runoff reduction, whereas that's not what you did: you measured cover rate and estimated runoff reduction. I would prefer that throughout you refer to 'cover rate as proxy of runoff reduction' to make it very clear what was measured.

L145 'literature', spelling mistake

L148 superscript 2 in m2

L149/50 this does not make sense - please consider the wording.

L155 weighed, not weighted

L153 It sounds like you did this all on one day, which I presume you did not - that would be a lot of work! Can you please say how long it took to sample (days, weeks?) and how you controlled for the fact that a later sample would mean the intercrops had longer to grow (and therefore would have more biomass)?

L158 How did you know that species sampled represented 80% of biomass in each quadrat, without harvesting it all and checking? I presume this is species that were *estimated* to be above 80%.

L173-5 It might be worth adding a brief explanatory sentence as to why you did this cutting and mixing - was it to control for something, or for some other reason? It feels oddly labour-intensive.

L185-6 I don't understand this sentence. What does 'with the ratio' mean?

L195 I don't understand 'weighted so as non-scanned samples' - do you mean 'weighed as non-scanned samples'?

L200-1 Why didn't you use your own measurement of Depth80 here? Feels like that would be better for your own experiment rather than relying on Garcia 2020?

L211 It's unclear what you mean by 'the amount of organic matter restitutions'.

L221 ref 51 is simply a quote of ref 60 so I would remove ref 51 from here. Reference 51 in the list of references also has a curious 'cited' date which should be removed.

L237 I'm not sure 'pic' is the right word.

L254 weighed, not weighted (check all other instances of 'weighted' - I'm not going to list any more)

L259 eq2 WS can't have the units mm if BD has g and cm-3 in it?

L261 superscript -3

L341 use p, not rho, same line 375-379.

Table 2: Why not show correlation coefficients for all entries in this table? I get that you want to show the significant correlations, but I don't see why you can't show all of them. Same with figure 2.

Table 4 Asteraceae is spelt wrong. Plantaginae should be Plantaginaceae.

Throughout, use . not , as a decimal point indicator (sorry, American formatting)

L409 this is figure 4.

L416 you say Nitrogen Supply was very close to 0 for all clusters, but then say it was significantly different between clusters 2-1 and 3-1. This feels contradictory?

L423 This is figure 5.

L431 I'm not sure you can say this vegetation was spontaneous, given you seeded it?

L439 should 'assumption' be 'hypothesis'?

L444 change 'were' to 'have been' to make it sound less like this proof came from the current study.

L464-470 don't seem to add anything to the discussion, but instead just restate what you did in the study.

L472 I'm not convinced that this study does emphasise this need.

L476 'Bunches' is a curious word to choose. Maybe 'combinations'?

L520 What is 'soluble concentration'? Do you mean 'solute concentration'?

L521: similarly, what is 'root water-soluble concentration'?

L555 this is figure 6. (Update numbers within text too on eg L574) I find the acronyms in this make it almost incomprehensible. It feels like there is enough space to fully spell out what each of the acronyms mean.

L608-611 can be deleted.

Reviewer #4: Dear authors, I congratulate with you for your very interesting and innovative study. I think you responded extensively to the reviewers’ comments and improved a lot the paper. I have one general comment and few minor amendments to highlight.

The general comment is about the number of references, that I find too high for a research article. Please try to reduce a bit their number, if possible.

Detailed comments follow here:

L212: please report if 10 g each are dry matter or fresh matter

L215: “40C” should be “40°C”

L217-218: “adjacent sieves”. Do you mean the one with the immediately lower and immediately higher mesh size?

L240 and L242: why 103°C and not 105°C that I suppose was the standard oven-drying temperature for gravimetric soil humidity determination?

L348-350: please double check because it seems that the trends were wrongly reported. What “only slightly to Water Supply” does it mean? “The relation was positive…”: seems it is the opposite

L359: add also Run-off reduction after Biomass Production

Table 4: are there any reasons why in the post hoc test column of cluster 2 the statistical letters are reported as capital letters?

Table 4 footnote: after “Different letters indicate significant differences” I would add “between clusters. “striking characteristics” should be explain better (e.g., in brackets you could report something like “i.e., characteristics for which the value of the cluster is significantly higher or lower from all the other two clusters”)

L391: it is “Figure 4” and not 3

L405: it is “Figure 5” and not 4

L438: Mg ha-1 of dry matter?

L462: replace commas with dots in 0,58 and 0,49

L500: consider to replace “homogeneous plot” with “homogeneous field”

L531: it is “Figure 6” and not 5

L540-541: please check the presence of too many spaces between words

L585-586: sounds like a repetition of the first sentence

L595: I would suggest adding also climate to cropping systems and soils

7. PLOS authors have the option to publish the peer review history of their article (what does this mean?). If published, this will include your full peer review and any attached files.

Reviewer #3: No

Reviewer #4: **Yes:** Daniele Antichi

---

## [Author Response · Author response to Decision Letter 2]

27 Jan 2026

We provide a specific file containing detailed responses to all the reviewers' comments.

---

## [Editor Report · Decision Letter 2]

1 Feb 2026

Assessing the Multifunctionality of Service Crops in Mediterranean Vineyards Using a Functional Trait Approach

PONE-D-25-16076R2

Dear Dr. Damour,

We’re pleased to inform you that your manuscript has been judged scientifically suitable for publication and will be formally accepted for publication once it meets all outstanding technical requirements.

Kind regards,

Raed Abduljabbar Haleem, Ph.D

Academic Editor

PLOS One

---

## [Editor Report · Acceptance letter]

PONE-D-25-16076R2

PLOS One

Dear Dr. Damour,

I'm pleased to inform you that your manuscript has been deemed suitable for publication in PLOS One. Congratulations! Your manuscript is now being handed over to our production team.

Kind regards,

on behalf of

Dr. Raed Abduljabbar Haleem

Academic Editor

PLOS One